# Comparing Reference Evapotranspiration Calculated in ETo Calculator (Ukraine) Mobile App with the Estimated by Standard FAO-Based Approach

**Pavlo Lykhovyd** 

The Institute of Irrigated Agriculture of the National Academy of Agrarian Sciences,
73483 Kherson, Naddniprianske, Ukraine; pavel.likhovid@gmail.com; Tel.: +380-660629897

**Abstract:** Reference evapotranspiration (ETo) is a key agrometeorological index for rational irrigation management. The standard method for ETo estimation, proposed by the FAO, is based on a complicated Penman–Monteith equation and requires many meteorological inputs, making it difficult for practical use by farmers. At present, there are many alternative simplified approaches for ETo estimation; most of them are directed at cutting the number of required meteorological inputs for calculation. Among them, special attention should be paid to the various temperature-based methods of ETo assessment. One of the temperature-based models for ETo computation was realized in the free mobile app ETo Calculator (Ukraine). The app gives Ukrainian farmers an opportunity to assess ETo values on a daily or monthly scale using mean air temperature, obtained through free online meteorological forecasts and archive services, as the only input. The objective of the study was to test the app's accuracy compared to FAO-based calculations in five key regions of Ukraine, each representing a particular climatic zone of the country. It was established that the app provides relatively good accuracy of ETo estimation even in raw (not adjusted to wind speed and relative air humidity) runs. The results of the statistical comparison with the FAO-calculated values on the daily scale are as follows: $R^2$ within 0.82–0.87, RMSE within 0.74–0.81 mm, MAE within 0.60–0.70 mm, MAPE within 18.07–25.50%, depending on the region. The results of the statistical comparison with the FAO-calculated values on the monthly scale are: $R^2$ within 0.88–0.95, RMSE within 0.50–0.72 mm, MAE within 0.33–0.59 mm, MAPE within 8.96–24.08% depending on the region. The ETo Calculator (Ukraine) is a good alternative to the complicated Penman–Monteith method and could be recommended for Ukrainian farmers to be used for irrigation management.

**Keywords:** agrometeorology; evapotranspiration; mobile application; regression analysis; water management

## 1. Introduction

The rational use of water resources in agriculture is crucial for ecological sustainability. Current scientific evidence supports the idea of an increasing global scarcity of fresh water with a simultaneous deterioration of water quality. Agriculture is one of the most demanding branches of freshwater in the global economy, and the demand is expected to increase in the near future due to the aggravation of global warming, especially in vulnerable areas of Africa, Middle East, and Southern Asia. Ukraine, especially its southern regions, is also facing the problem of scarcity and low quality of irrigation water due to the aggravation of the processes of aridity increasing in most areas of intense crop production. Even the western regions of the country, which used to have a sufficient natural moisture supply for crop production, are suffering from increased dryness and a moisture deficit [1–3]. Therefore, the problem of rational and economical use of water in the agricultural sector, especially in the field of crop production, is relevant for modern society.

Rational irrigation water management is impossible without a scientifically based approach to determine the water demands for crop production. The basis for this is the

calculation of reference evapotranspiration—an index which represents the loss of water over a certain time span from a certain land plot covered with a grass surface (height of 0.12 m; fixed surface resistance of 70 s/m; albedo of 0.23) that is well watered [4]. Reference evapotranspiration (or ETo) is a basic parameter for the further estimation of the irrigation requirements for a particular crop. Therefore, an accurate and operational estimation of ETo is crucial to establish irrigation demands and irrigation scheduling [5].

Currently, dozens of different approaches for evapotranspiration assessment have been developed, while the best one in terms of accuracy is direct field measurement using a lysimeter; however, it is expensive, laborious, and unsuitable for production conditions. Considering the mentioned drawbacks, indirect methods for reference evapotranspiration derivation from meteorological data were developed [6–8]. Each computation approach has its own unique algorithm and advantages, but finally the FAO and the scientific society approved the Penman–Monteith equation as the standard method [9]. This methodology was used in most of the software for determining irrigation demands, both FAO-delivered (as CROPWAT, AquaCrop, etc.) or provided by exterior developers. The main weakness of this method is its high complexity and demand for a large number of meteorological inputs, which are often inaccessible to the ordinary farmer. In addition, there is a lack of free mobile apps for ETo estimation using the Penman–Monteith equation that makes it unsuitable for use in field conditions. Hence the necessity for a simplification of the reference evapotranspiration calculation model arose, and most developers struggle to cut the number of required meteorological inputs without loss of estimation accuracy [10,11]. For example, Hargreaves and Priestley-Taylor methods are used for the ETo evaluation using limited meteorological inputs. These methods are less popular than the Penman–Monteith method and are less popular among scientists, although they provide a reasonably high precision in evaluating reference evapotranspiration [12,13]. On the other hand, fully automated computation mobile systems were developed, e.g., EVAPO and AgSAT mobile apps, requiring just the coordinates of the irrigation plot on the global map for an automatic estimation of ETo using external data from NASA servers [14]. This approach is quite comfortable for the user, but the studies found that it is not reliable enough [15,16].

Further, another approach for ETo assessment on the local level was proposed: temperature-based regression models. For example, such models were developed for every region of Ukraine and then aggregated in the mobile app ETo Calculator (Ukraine) [17,18]. Although the approach is promising, it remains unclear whether the developed models are reliable and accurate in reference evapotranspiration estimation using meteorological data that do not fall within the period of 1971–2020 (the period used to create models for ETo assessment). The goal of this study was to evaluate the performance of the ETo Calculator (Ukraine) mobile app in the estimation of reference evapotranspiration in 2021 by the key regions of the country in comparison to the estimation using the FAO-based calculations of the Penman–Monteith equation.

## 2. Materials and Methods

The assessment of the accuracy of ETo Calculator (Ukraine) was performed through direct comparison of its calculations with those performed in an FAO-based add-in for MS Excel. The mobile app is available for download and installation on Android smartphones via the link https://play.google.com/store/apps/details?id=com.EvapUkr (the accessed date is 20 June 2022). The calculations by Penman–Monteith were made using an adapted ETo assessment tool for MS Excel developed by Sherzod Rusmetov (guidelines and download link are available for free at https://youtu.be/1xT1CmDe2gc, the accessed date is 20 June 2022), and engaged such inputs as site elevation, latitude, minimum and maximum air temperature, windspeed, sunlight hours, etc. Missing meteorological data were estimated as recommended by FAO [19]. Meteorological data were taken from the observations of regional hydrometeorological centers (available at http://pogodaiklimat.ru, the last accessed date 20 June 2022) and meteorological reference books [20,21]. The results

obtained within the ETo Calculator (Ukraine) were not adjusted as in the app guidelines, but we took the mean reference evapotranspiration values computed in the app.

The study was conducted for the period with mean daily air temperature above zero in 2021 (precondition for successful use of ETo Calculator (Ukraine), which can compute reference evapotranspiration only if air temperature is above zero). There were 322 such days in Kherson oblast, 321 in Mykolaiv, 300 in Dnipropetrovsk, 303 in Cherkasy, 299 in Chernihiv, and 325 in Uzhhorod (Zakarpattia), respectively. Each region was chosen to represent the general climatic conditions of different zones of the country: Kherson—dry steppe zone; Mykolaiv—southern moderately dry steppe zone; Dnipro—northern steppe zone; Cherkasy—forest steppe zone; Chernihiv—Polissia; Uzhhorod (Zakarpattia)—forest zone. The zoning of the Ukrainian territory was taken according to the study [22]. The schematic image of the study location is presented in the Figure 1.

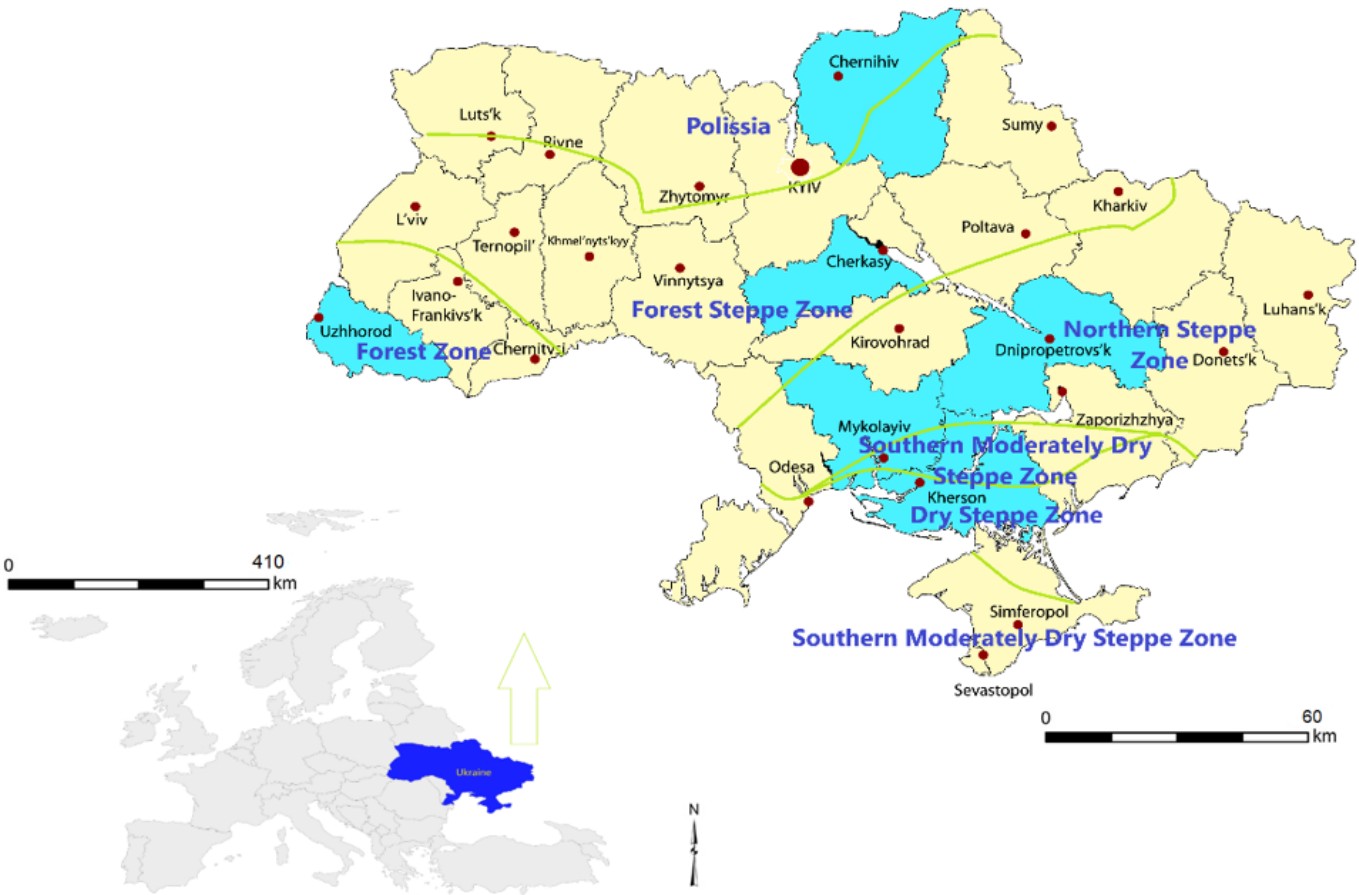

**Figure 1.** Location of the study regions on the map of Europe and the map of Ukraine (the studied territories are marked with turquoise color; green lines are the borders of the climate zones).

In addition, a monthly comparison for 2021 was performed to assess the app performance more comprehensively. There were 10, 9, 9, 9, 9, and 11 comparison pairs in this case for Kherson, Mykolaiv, Dnipropetrovsk, Cherkasy, Chernihiv, and Uzhhorod, respectively.

The computation of reference evapotranspiration in the app is carried out using following individual Equations (1)–(5) for each studied region:

$$Kherson = 0.2473 \times T \tag{1}$$

$$Mykolaiv = 0.2424 \times T \tag{2}$$

$$Dnipropetrovsk = 0.2609 \times T \tag{3}$$

$$Cherkasy = 0.2413 \times T \tag{4}$$

$$Chernihiv = 0.2461 \times T \tag{5}$$

where: T—average daily air temperature, °C.

The Equations (1)–(5) were derived as a result of regression analysis on the interrelation of regional reference evapotranspiration and average air temperature using the long-term data set of 1970–2020.

Statistical comparison included evaluation of indices such as *R*, $R^2$, RMSE, MAE, and MAPE. Correlation and determination coefficients were calculated to evaluate the quality of the fit, while errors were assessed to evaluate the general accuracy of the app performance. Common procedures were used to calculate the statistical parameters using MS Excel 365 table processor [23–28]. The closer $R^2$ values are to 1.00, the better the model fitting quality [29]. The MAPE values were interpreted using the guidelines by Blasco et al. [30]. The following Equations (6)–(11) were utilized in the statistical calculations:

$$R_{XY} = \frac{\sum_1^N (x_i - \overline{x})(y_i - \overline{y})}{(n-1)s_X s_Y} \tag{6}$$

where: $s_X$, $s_Y$—standard deviations for *X* and *Y*, respectively; *n*—number of data; $x_i$, $y_i$—values of the studied parameters in the pair; $\overline{x}$, $\overline{y}$ —mean values for *X* and *Y*, respectively.

$$MSE = \frac{1}{n} \sum_{i=1}^n (Y_i - \hat{Y}_i)^2 \tag{7}$$

where: *n*—number of data; $Y_i - \hat{Y}_i$—difference between the observed and predicted values.

$$RMSE = \sqrt[2]{MSE} \tag{8}$$

$$MAPE = \frac{100\%}{n} \sum_{t=1}^n \left| \frac{Y_i - \hat{Y}_i}{Y_i} \right| \tag{9}$$

$$MAE = \frac{\sum_{i=1}^n |e_i|}{n} \tag{10}$$

$$|e_i| = \left| Y_i - \hat{Y}_i \right| \tag{11}$$

Visual approximation was performed in MS Excel 365 to assist in visual assessment of ETo estimation accuracy and simplify the process of finding the pairs with the highest discrepancy in the index computation.

## 3. Results

The statistical comparison between Penman–Monteith and the temperature-based method for ETo assessment demonstrated a relatively good performance of the latter. The values of the correlation and determination coefficients for all the locations studied demonstrated a high quality of fitting (Table 1). Although RMSE and MAE values were relatively high (exceeding the water amount for a single watering through drip irrigation, which is taken as 0.50 mm in Ukraine), one must admit that they were lower than in the test run of alternative mobile app EVAPO with average RMSE of 0.95 mm [11]. Moreover, it should be stressed that the previously quoted estimation was performed on a raw run without previous adjustment of the ETo Calculator (Ukraine) computations to relative air humidity and windspeed.

**Table 1.** Statistical indices of ETo Calculator (Ukraine) mobile app accuracy comparing to Penman–Monteith method of reference evapotranspiration assessment (daily scale).

| Statistical Index | Region of Ukraine | | | | | |
| --- | --- | --- | --- | --- | --- | --- |
| | Kherson | Mykolaiv | Dnipropetrovsk | Cherkasy | Chernihiv | Uzhhorod (Zakarpattia) |
| $n$ | 322 | 321 | 300 | 303 | 299 | 325 |
| R | 0.93 | 0.93 | 0.92 | 0.91 | 0.92 | 0.91 |
| $R^2$ | 0.86 | 0.87 | 0.84 | 0.83 | 0.86 | 0.82 |
| RMSE (mm/day) | 0.75 | 0.74 | 0.81 | 0.80 | 0.74 | 0.77 |
| MAE (mm/day) | 0.61 | 0.60 | 0.70 | 0.64 | 0.62 | 0.63 |
| MAPE (%) | 18.58 | 18.07 | 20.69 | 20.86 | 22.22 | 25.50 |

A visual approximation of the ETo Calculator (Ukraine) calculations is presented in the Figure 2. It is evident that the highest discrepancy between the studied methods of reference evapotranspiration assessment was observed in the Cherkasy region, while the lowest ones were in Mykolaiv and Kherson (the driest regions among those studied). This fact tells us that there is an influential geographical component in the model of ETo Calculator (Ukraine) owing to which the accuracy of ETo estimation may vary significantly between the regions of the country.

To sum up, the best performance of the tested mobile app for daily scale estimation was recorded for Mykolaiv oblast of Ukraine (southern moderately dry steppe zone), while the worst performance was associated with Dnipropetrovsk oblast (northern steppe zone) because of the highest values of deviations RMSE and MAE.

Comparison on the monthly scale for 2021 was more optimistic with better fitting quality and considerably higher accuracy in all the studied regions (Table 2). A better fit of the model was obvious in the case of graphical comparison (Figure 3). Therefore, the ETo Calculator app is better for annual evapotranspiration rates than operational irrigation scheduling. The best performance was recorded for the Mykolaiv oblast, while the worst was in the Zakarpattia region.

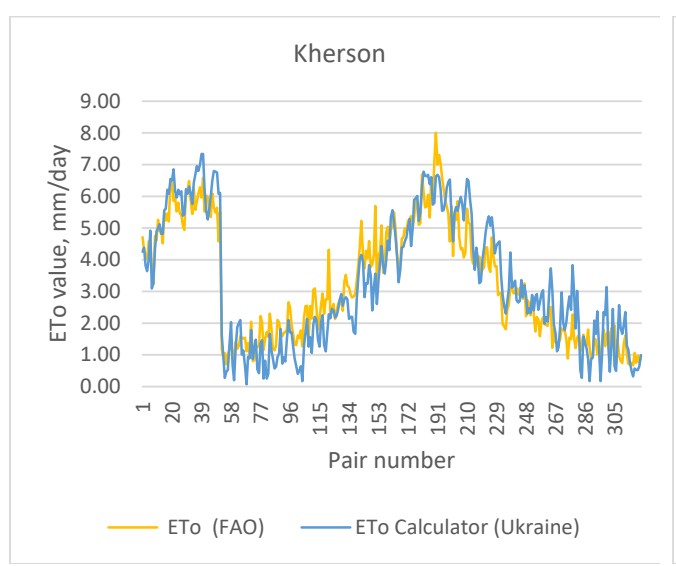 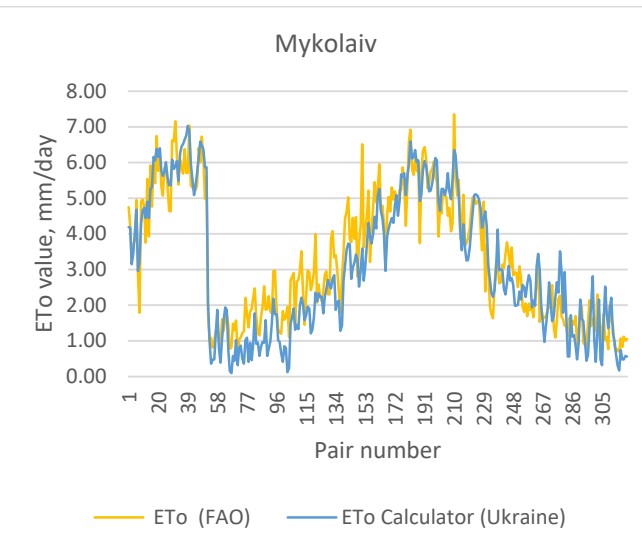

**Figure 2.** *Cont.*

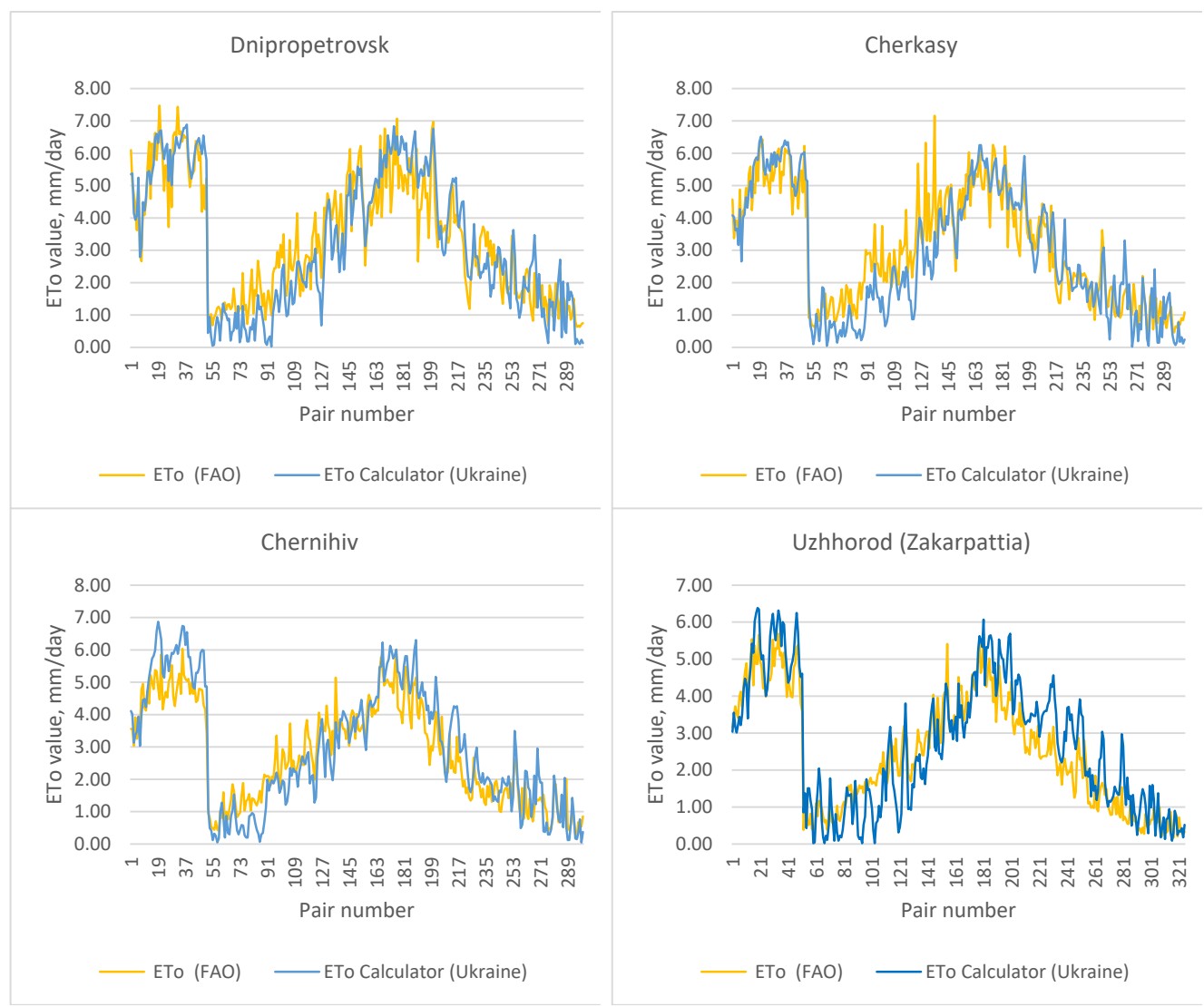

**Figure 2.** Reference evapotranspiration (mm/day) comparison between the standard FAO Penman–Monteith (Sherzod Rusmetov adaptation) and ETo Calculator (Ukraine) method by the studied regions of Ukraine (daily scale comparison).

**Table 2.** Statistical indices of ETo Calculator (Ukraine) mobile app accuracy compared to Penman–Monteith method of reference evapotranspiration assessment (monthly scale).

| Statistical Index | Region of Ukraine | | | | | |
|---|---|---|---|---|---|---|
| | **Kherson** | **Mykolaiv** | **Dnipropetrovsk** | **Cherkasy** | **Chernihiv** | **Uzhhorod (Zakarpattia)** |
| n | 10 | 9 | 9 | 9 | 9 | 11 |
| R | 0.96 | 0.98 | 0.97 | 0.97 | 0.97 | 0.94 |
| $R^2$ | 0.93 | 0.95 | 0.95 | 0.94 | 0.94 | 0.88 |
| RMSE (mm/day) | 0.61 | 0.50 | 0.57 | 0.57 | 0.56 | 0.72 |
| MAE (mm/day) | 0.51 | 0.33 | 0.47 | 0.46 | 0.50 | 0.59 |
| MAPE (%) | 15.04 | 8.96 | 12.90 | 13.45 | 16.81 | 24.08 |

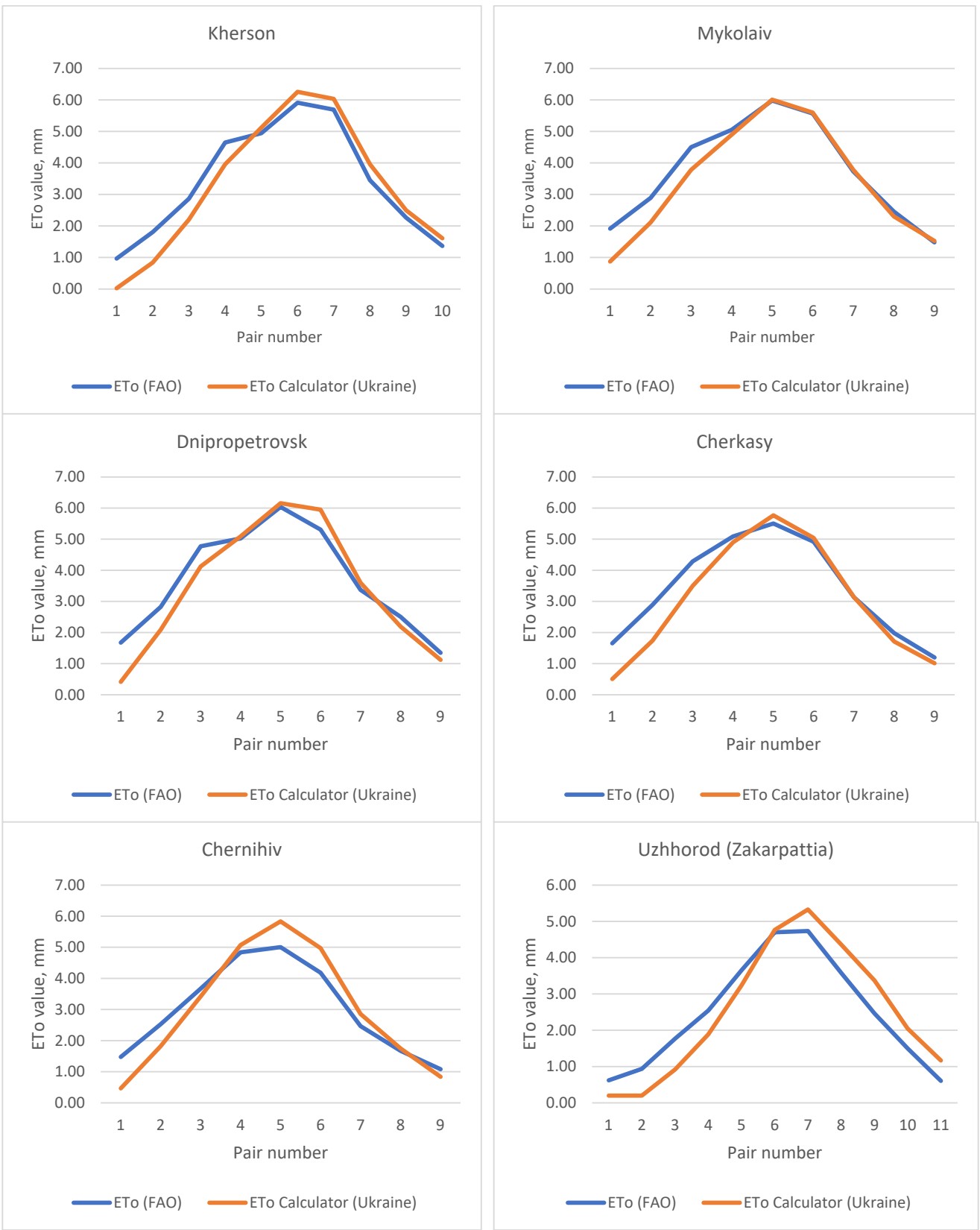

**Figure 3.** Reference evapotranspiration (mm/day) comparison between the standard FAO Penman–Monteith (Sherzod Rusmetov adaptation) and ETo Calculator (Ukraine) method by the studied regions of Ukraine (monthly scale comparison).

## 4. Discussion

Although the direct lysimetric method is indubitably superior to all other indirect methods of evapotranspiration assessment, there is a lack of available lysimetric stations, and therefore, it is used primarily for scientific research purposes for the validation and calibration of the computation methods for the estimation of evapotranspiration [31–33]. For example, the standard FAO method of the Penman–Monteith equation has been validated and calibrated using lysimetric data; therefore, its performance is considered the best one among the other indirect computational approaches for assessing ETo [34,35].

However, the requirement of many meteorological inputs, which are rarely accessible to Ukrainian farmers, limits the practical implementation of the Penman–Monteith-based assessment of reference evapotranspiration. The quest to cut the number of inputs resulted in the development of alternative approaches for ETo calculation, the most popular among which are the Hargreaves and Makkink equations [36–38]. The above methods use more common weather data from meteorological stations; however, some data (e.g., amount of extra-terrestrial radiation) are still inaccessible for certain territories due to the lack of weather stations and specific recording equipment [39]. Mean air temperature remains the easiest meteorological index to obtain through direct measurement in the field or even using weather data provided by forecast services. Therefore, reference evapotranspiration calculations based on air temperature have the highest prospect for practical implementation by farmers.

At the same time, new simplified approaches must be accurate. However, even the standard Penman–Monteith approach fails to provide reliable reference evapotranspiration assessment in some environmental conditions without previous calibration [40]. Therefore, high accuracy and reliability is desirable, but it might strongly depend on the environment. Thus, it is difficult to develop one tool for ETo assessment in any region of the planet with similar efficiency in different climatic zones. The ETo Calculator (Ukraine) takes this fact into account and proposes different models for the estimation of reference evapotranspiration even within the borders of one country. Of course, it could be considered a drawback that the app is suitable just for in-Ukraine calculations, but this grants the highest possible accuracy of calculations in the zones, which are embraced in the mobile app. In addition, we cannot hold back the fact that the models used as the basis for reference evapotranspiration assessment in the studied app have their imperfections, mainly connected with the linear regression approach used in the process of development. The quality of the models could be improved by the implementation of an artificial neural network (ANN) approach to data analysis or a combined "regression—ANN" approach to prediction as it was shown to be superior to separate regression or ANN-based methods in agricultural modeling [41]. Now, it seems that regression-based models used in the ETo Calculator (Ukraine) are inferior to such simplified ETo assessment methods as the Hargreaves or Priestley–Taylor method, although the ETo Calculator (Ukraine) calculations are much simpler for an ordinary farmer. In addition, it is evident that the daily results of the reference evapotranspiration assessment require more careful calibration and adjustment compared to the results of the monthly index assessment. Furthermore, it should be noted that there is a great discrepancy in the accuracy of the app for different zones of Ukraine: the best performance being in the southern regions, while the worst is in the West and the North. This issue also requires addressing in further app updates.

There are some other software options for ETo assessment. For example, computer-based programs produced by FAO Eto Calculator, CROPWAT (perhaps, the most popular and reliable ones, but unsuitable for field conditions because of the absence of mobile apps). If FAO software is too complicated, one can use software based on alternative calculation methods such as DailyET [42] or one of the latest developments in this field for calculations using limited weather data [11]. However, more meteorological inputs are needed to perform the ETo assessment in the mentioned above software. In addition, reference evapotranspiration forecasts for the near future are possible within the framework of FORETo ANN-based software, although it should be noted that at the moment its precision

is not suitable for irrigation scheduling and planning due to the high errors, RMSE of 0.98 mm [43].

Although all the above-mentioned software has its advantages, the common drawback is that none of it is portable and each requires a PC for calculations, which is not always suitable because of the low level of provision of farmers with laptops that have Internet access, while smartphones are available for most Ukrainian farmers. In this regard, ETo Calculator (Ukraine) is the only mobile app product with a relatively high accuracy of operational reference evapotranspiration estimation for Ukrainian farmers.

## 5. Conclusions

The results of the comparative statistical analysis between ETo calculated using the Penman–Monteith method and the simplified computations in the ETo Calculator (Ukraine) mobile app testify that the in-app computations are less accurate than those conducted by the standard method, although the accuracy is on a reasonable level. The ETo Calculator (Ukraine) mobile app in a raw run could not be recommended for precise irrigation scheduling because of the high likelihood of significant error occurrence. At the same time, monthly ETo assessment provides a much better prediction of reference evapotranspiration; therefore, it is advisable to use the app for a rough evaluation of ETo dynamics on a seasonal or annual scale for a better understanding of the field water balance dynamics. Further improvements in the computation methodology are required. It is highly likely that the introduction of an ANN-based approach will enhance the quality of ETo assessment in the app.

**Funding:** This research received no external funding.

**Institutional Review Board Statement:** Not applicable.

**Informed Consent Statement:** Not applicable.

**Data Availability Statement:** Not applicable.

**Conflicts of Interest:** The author declares no conflict of interest.

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
