# Peer review of "Comparing Reference Evapotranspiration Calculated in ETo Calculator (Ukraine) Mobile App with the Estimated by Standard FAO-Based Approach"

_agriengineering, doi:10.3390/agriengineering4030048_

Round 1
Reviewer 1 Report
What is the method/equation used in ETo calculator app? Please provide the details.
Abstract
L20: “to windspeed and relative air humidity) run; the results of statistical comparison” to “to windspeed and relative air humidity) run. The results of statistical comparison”
Introduction
L37: “to the determination of water demands for crop production.” To “to determine the water demands for crop production.”
L44: Lysimeter measures ETc not ETo. Please correct the statement.
L48-49: “but finally FAO and scientific society approved as the standard the method known as Penman-Monteith equation [6]. This methodology was put in the basis for most software used to determine irrigation demands” to “but finally FAO and scientific society approved the Penman-Monteith equation as the standard method [6]. This methodology was used in most of the software for determination of irrigation demands”
L53: “which are often inaccessible for ordinary farmer.” To “which are often inaccessible to ordinary farmer.”
L59-60: “the global map and waiting for automatic estimation of ETo by the external data” to “the global map for automatic estimation of ETo using the external data”
L66: “The approach is promising; however, it remains” to “Though the approach is promising, it remains”
L83-84: “Calculations results” to “The results”
L96: “evaluation of such indices as R” to “evaluation of indices like R”
Please try to maintain uniformity of symbols used for real and forecasted values in the equations. I think it should be predicted values not forecasted values. Forecast performs prediction into the future, in a time range beyond the last instant of measured data. In contrast, the predict command predicts the response of an identified model over the time span of measured data (Source: https://in.mathworks.com/help/ident/ref/idmodel.forecast.html#:~:text=forecast%20performs%20prediction%20into%20the,time%20span%20of%20measured%20data.)
Results section can be improved by comparing performance of the mobile app with FAO model at daily and monthly scale. At present it is too shallow. Accordingly, discussion can be improved.
Author Response
Dear Reviewer!
Thank you for your time and efforts directed to enhance the value and quality of the manuscript. I will try to satisfy your reasonable remarks.
What is the method/equation used in ETo calculator app? Please provide the details
The equations used in the mobile app are provided in the section Materials and Methods.
Abstract
L20: “to windspeed and relative air humidity) run; the results of statistical comparison” to “to windspeed and relative air humidity) run. The results of statistical comparison”
Corrected in the manuscript body.
Introduction
L37: “to the determination of water demands for crop production.” To “to determine the water demands for crop production.”
L44: Lysimeter measures ETc not ETo. Please correct the statement.
L48-49: “but finally FAO and scientific society approved as the standard the method known as Penman-Monteith equation [6]. This methodology was put in the basis for most software used to determine irrigation demands” to “but finally FAO and scientific society approved the Penman-Monteith equation as the standard method [6]. This methodology was used in most of the software for determination of irrigation demands”
L53: “which are often inaccessible for ordinary farmer.” To “which are often inaccessible to ordinary farmer.”
L59-60: “the global map and waiting for automatic estimation of ETo by the external data” to “the global map for automatic estimation of ETo using the external data”
L66: “The approach is promising; however, it remains” to “Though the approach is promising, it remains”
L83-84: “Calculations results” to “The results”
L96: “evaluation of such indices as R” to “evaluation of indices like R”
Corrected in the manuscript body.
Please try to maintain uniformity of symbols used for real and forecasted values in the equations.
The equations are corrected.
I think it should be predicted values not forecasted values. Forecast performs prediction into the future, in a time range beyond the last instant of measured data. In contrast, the predict command predicts the response of an identified model over the time span of measured data (Source: https://in.mathworks.com/help/ident/ref/idmodel.forecast.html#:~:text=forecast%20performs%20prediction%20into%20the,time%20span%20of%20measured%20data.)
You are right. The term was substituted.
Results section can be improved by comparing performance of the mobile app with FAO model at daily and monthly scale. At present it is too shallow. Accordingly, discussion can be improved.
The first version of the manuscript contained the generalized results of the daily comparison. Monthly comparison results were added (see Table 2 and the Figure 3), and discussion section was extended.
Corrections made in the manuscript body, are marked yellow; however, some of them because of the coincidence with the corrections made with accordance to other reviewers, might fall under another color.
In addition, the manuscript has undergone a grammar revision, but I hope that the language corrections made did not spoil the paper.
Thank you very much for your valuable remarks!
Sincerely
Dr Pavlo Lykhovyd

Author Response

(The authors gave the same response as above.)

Reviewer 3 Report
Reference evapotranspiration (ETo) estimation plays important role in agriculture in many different ways. In the presented study’s case ETo estimation based on air temperature is compared to the values from Penman-Monteith equation. The topic is very important for research and development of various tools for the ETo prediction purpose. For broader audience of agro meteorologists and agronomists as well as for those who are interested in digital tool development for climate services the presented topic would be very attractive. Introduction in general is written well. Methodology is lacking several details. Good analysis of the results and discussion are also provided. The manuscript has easy to follow narrative but needs English editor revision. Therefore, my recommendation is that the manuscript can be accepted after major revisions that can be easily implemented and could improve it.
Here are my several concerns/suggestions I think to be addressed by authors:
1. I think the title is a little misleading. The content of the manuscript is not comparison of the two methods, but author compares his estimates to those calculated based on Penman-Monteith equation. There is not data against which both approaches were compared. Author claims in L17, that “the goal of the study was to test the app accuracy comparing to FAO-based calculations”. Thus, the title needs to be changed.
2. Regression approach has its weakness that is predicting outside its limits by extrapolating. Models were derived 1970-2020 years data. Could you provide justification using only one year data for evaluation.
3. As temperature-based approach is the base for the results provided in sec 2: “materials and methods” add a paragraph that provides basics how regressions were derived, including observed data sources on temperature and ET0.
4. I would recommend adding 1:1 line as Fig. 3, in addition to existing Fig. 2 for better illustration.
Here are some more specific comments:
L6: provide more information here: city, country, what NAAS is.
L10: edit “great number”. It is a little exaggeration.
L16: “mean air temperature”: is this daily or monthly? How can farmers get it?
L29-30: revise this sentence.
L33-35: revise this sentence, also to come to this conclusion include Ukraine in your narrative above, as your application can be used only for that region.
L43: add references here.
L47-61: there is also Priestly-Taylor approach that uses less inputs (not required wind speed, air humidity) that should be also discussed and referenced here.
Fig. 1 : swap two maps: put main map with regions in the center and map of Europe with Ukraine in the upper corner.
L 98: too many references for standard statistics.
L100-111: remove those statistical parameters that are not discussed later, and results are not provided in the tables.
L100-111: instead of “real value” I would recommend using “observed”.
L124: four
Fig. 2: what is “Pair number” here not clear from the text. Also missing units for ETo on vertical axes.
In Table 1: added number of samples (N) used in calculations.
L188-191. revise this sentence.
L188-198: please mention and discuss here weakness of the regression model approach you used here.
L207-208: I do not think that PC requirements is too much limitation here considering different modifications of laptops.
213-214: you can only judge here how your results are close to those of Penman-Monteith formula results.
Author Response
Dear Reviewer!
Thank you very much for your time and efforts to evaluate the quality of the manuscript to make it better and enhance its scientific value.
I tried to address your remarks and concerns, let me give you answers to your questions.
- I think the title is a little misleading. The content of the manuscript is not comparison of the two methods, but author compares his estimates to those calculated based on Penman-Monteith equation. There is not data against which both approaches were compared. Author claims in L17, that “the goal of the study was to test the app accuracy comparing to FAO-based calculations”. Thus, the title needs to be changed.
I tried to change the title so that it will better emphasize the main idea of the study. Please, evaluate the new version.
- Regression approach has its weakness that is predicting outside its limits by extrapolating. Models were derived 1970-2020 years data. Could you provide justification using only one year data for evaluation.
You are absolutely right about regression models. However, in my humble opinion, it is not reasonable to include the period of model training (any year of the 1970-2020 time span) into the assessment, it is better to give a trial for the model using the data that were not used during its creation. And at current moment, in my disposal are just data of 2021.
- As temperature-based approach is the base for the results provided in sec 2: “materials and methods” add a paragraph that provides basics how regressions were derived, including observed data sources on temperature and ET0.
I added the equations of models for each region. The meteorological data sources are listed in the section. They are available on the website http://pogodaiklimat.ru, as well as in the Ukrainian meteorological reference books quoted, namely [15–16]:
Lipinsky, V.M.; Diachuk, V.A.; Babichenko, V.M. Climate of Ukraine; Rayevsky Publishing House: Kyiv, Ukraine, 2003.
Galik, O.I.; Basiuk, T.O. Guidelines “Reference book on Ukrainian climate” for conduction of practical, computation, diploma and master’s degree work for the students of full and correspondent education forms of life sciences specialties of the NUWEEM; Rivne, Ukraine, 2014
- I would recommend adding 1:1 line as Fig. 3, in addition to existing Fig. 2 for better illustration.
I am very sorry, but I failed to understand properly what you meant by ‘adding 1:1 line as Fig. 3”.
L6: provide more information here: city, country, what NAAS is.
The details were added.
L10: edit “great number”. It is a little exaggeration.
Changed to “many”.
L16: “mean air temperature”: is this daily or monthly? How can farmers get it?
The mean air temperature is daily (if assessed on the daily scale) or monthly (for the monthly scale, respectively). It is accessible from any internet-based meteorological service, for example, https://rp5.ru or any other (they are numerous, including https://meteopost.com, https://pogodaiklimat.ru, etc.)
L29-30: revise this sentence.
L33-35: revise this sentence, also to come to this conclusion include Ukraine in your narrative above, as your application can be used only for that region.
The sentences were revised, and the information you requested were added together with the references.
L43: add references here.
This section was totally rewritten according to the remarks of all the Reviewers, so, it is completely changed.
L47-61: there is also Priestly-Taylor approach that uses less inputs (not required wind speed, air humidity) that should be also discussed and referenced here.
The requested information was added together with required references.
Fig. 1 : swap two maps: put main map with regions in the center and map of Europe with Ukraine in the upper corner.
I tried to do my best in order to substitute the maps. I hope that the revised Figure 1 looks good.
L 98: too many references for standard statistics.
The problem is that some reviewers require the references to be placed for every statistical index used in the study, so I decided to leave this section as it is.
L100-111: remove those statistical parameters that are not discussed later, and results are not provided in the tables.
All the statistical indices listed are used in the study and presented in the Tables 1-2. If you mean MSE and R, these indices are used in calculations of RMSE and R2, therefore, it is required to provide them for proper explanation on calculations.
L100-111: instead of “real value” I would recommend using “observed”.
The term was changed as you recommended.
L124: four
The mistake is that it had to be ‘previously quoted’. The mistake was corrected.
Fig. 2: what is “Pair number” here not clear from the text. Also missing units for ETo on vertical axes.
The units for ETo were added in the Figure 2. Pair number means the number of a comparison pair “ETo FAO vs ETo Calculator (Ukraine) app”.
In Table 1: added number of samples (N) used in calculations.
The number of samples is given in the text of the Materials and Methods section. But I also added this line in the Tables 1-2.
L188-191. revise this sentence.
The sentence was revised and changed.
L188-198: please mention and discuss here weakness of the regression model approach you used here.
Brief discussion of the issue was added.
L207-208: I do not think that PC requirements is too much limitation here considering different modifications of laptops.
Additional explanation is given.
213-214: you can only judge here how your results are close to those of Penman-Monteith formula results
Yes, you are right, the Conclusions were changed.
The corrections were marked green, although, considering their mixing with the corrections made due to the suggestions of other reviewers, I would like to mention that some of them might fall under another color scheme. All the colored text in the manuscript is corrections. Please, if you have time, have a look through all the colored text for better understanding of the changes in the paper.
In addition, the manuscript has undergone a grammar revision, but I hope that the language corrections made did not spoil the paper.
Thank you very much for your valuable review comments and suggestions!
Sincerely
Dr Pavlo Lykhovyd

Round 2
Reviewer 1 Report
The authors have made substantial changes in the revised manuscript. My only comment is when they have converted daily data to monthly data, it should contain 12 months/values. But they have mentioned 10, 9, 9, 9, 9, and 11 comparison pairs. Please check it.
Reviewer 2 Report
The authors have made extensive revision to the paper. This paper is noe acceptable for publication.